# GENERATIVE MODELS FROM THE PERSPECTIVE OF CONTINUAL LEARNING

## ABSTRACT

Which generative model is the most suitable for Continual Learning? This paper aims at evaluating and comparing generative models on disjoint sequential image generation tasks. We investigate how several models learn and forget, considering various strategies: rehearsal, regularization, generative replay and fine-tuning. We used two quantitative metrics to estimate the generation quality and memory ability. We experiment with sequential tasks on three commonly used benchmarks for Continual Learning (MNIST, Fashion MNIST and CIFAR10). We found that among all models, the original GAN performs best and among Continual Learning strategies, generative replay outperforms all other methods. Even if we found satisfactory combinations on MNIST and Fashion MNIST, training generative models sequentially on CIFAR10 is particularly instable, and remains a challenge. Our code is anonymously available online [1].

## 1 INTRODUCTION

Learning in a continual fashion is a key aspect for cognitive development among biological species (Fagot & Cook, 2006). In Machine Learning, such learning scenario has been formalized as a Continual Learning (CL) setting (Srivastava et al., 2013; Nguyen et al., 2017; Seff et al., 2017; Shin et al., 2017; Schwarz et al., 2018). The goal of CL is to learn from a data distribution that change over time without forgetting crucial information. Unfortunately, neural networks trained with back-propagation are unable to retain previously learned information when the data distribution change, an infamous problem called "catastrophic forgetting" (French, 1999). Successful attempts at CL with neural networks have to overcome the inexorable forgetting happening when tasks change.

In this paper, we focus on generative models in Continual Learning scenarios. Previous work on CL has mainly focused on classification tasks (Kirkpatrick et al., 2017; Rebuffi et al., 2017; Shin et al., 2017; Schwarz et al., 2018). Traditional approaches are *regularization*, *rehearsal* and *architectural* strategies, as described in Section 2. However, discriminative and generative models strongly differ in their architecture and learning objective. Several methods developed for discriminative models are thus not directly extendable to the generative setting. Moreover, successful CL strategies for generative models can be used, via sample generation as detailed in the next section, to continually train discriminative models. Hence, studying the viability and success/failure modes of CL strategies for generative models is an important step towards a better understanding of generative models and Continual Learning in general.

We conduct a comparative study of generative models with different CL strategies. In our experiments, we sequentially learn generation tasks. We perform ten disjoint tasks, using commonly used benchmarks for CL: MNIST (LeCun et al., 1998), Fashion MNIST (Xiao et al., 2017) and CIFAR10 (Krizhevsky et al., 2009). In each task, the model gets a training set from one new class, and should learn to generate data from this class without forgetting what it learned in previous tasks, see Fig. 1 for an example with tasks on MNIST.

We evaluate several generative models: Variational Auto-Encoders (VAEs), Generative Adversarial Networks (GANs), their conditional variant (CVAE ans CGAN), Wasserstein GANs (WGANs) and

---

[1] https://github.com/anonymous-authors-2018/Generative_models_from_the_perspective_of_Continual_learning

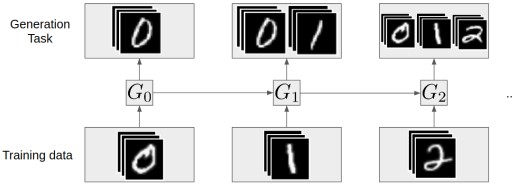

Figure 1: The disjoint setting considered. At task $i$ the training set includes images belonging to category $i$, and the task is to generate samples from all previously seen categories. Here MNIST is used as a visual example,but we experiment in the same way Fashion MNIST and CIFAR10.

Wasserstein GANs Gradient Penalty (WGAN-GP). We compare results on approaches taken from CL in a classification setting: *finetuning*, *rehearsal*, *regularization* and *generative replay*. *Generative replay* consists in using generated samples to maintain knowledge from previous tasks. All CL approaches are applicable to both variational and adversarial frameworks. We evaluate with two quantitative metrics, Fréchet Inception Distance (Heusel et al., 2017) and Fitting Capacity (Lesort et al., 2018), as well as visualization. Also, we discuss the data availability and scalability of CL strategies.

Our contributions are:

- Evaluating a wide range of generative models in a Continual Learning setting.
- Highlight success/failure modes of combinations of generative models and CL approaches.
- Comparing, in a CL setting, two evaluation metrics of generative models.

We describe related work in Section 2, and our approach in Section 3. We explain the experimental setup that implements our approach in Section 4. Finally, we present our results and discussion in Section 5 and 6, before concluding in Section 7.

## 2 RELATED WORK

### 2.1 CONTINUAL LEARNING FOR DISCRIMINATIVE MODELS

Continual Learning has mainly been applied to discriminative tasks. On this scenario, classification tasks are learned sequentially. At the end of the sequence the discriminative model should be able to solve all tasks. The naive method of fine-tuning from one task to the next one leads to catastrophic forgetting (French, 1999), i.e. the inability to keep initial performance on previous tasks. Previously proposed approaches can be classified into four main methods.

The first method, referred to as *rehearsal*, is to keep samples from previous tasks. The samples may then be used in different ways to overcome forgetting. The method can not be used in a scenario where data from previous tasks is not available, but it remains a competitive baseline (Rebuffi et al., 2017; Nguyen et al., 2017). Furthermore, the scalability of this method can also be questioned because the memory needed to store samples grows linearly with the number of tasks.

The second method employs *regularization*. Regularization constrains weight updates in order to maintain knowledge from previous tasks and thus avoid forgetting. Elastic Weight Consolidation (EWC) (Kirkpatrick et al., 2017) has become the standard method for this type of regularization. It estimates the weights' importance and adapt the regularization accordingly. Extensions of EWC have been proposed, such as online EWC (Schwarz et al., 2018). Another well known regularization method is *distillation*, which transfers previously learned knowledge to a new model. Initially proposed by Hinton et al. (2015), it has gained popularity in CL (Li & Hoiem, 2017; Rebuffi et al., 2017; Wu et al., 2018b; Shin et al., 2017) as it enables the model to learn about previous tasks and the current task at the same time.

The third method is the use of a *dynamic architecture* to maintain past knowledge and learn new information. Remarkable approaches that implement this method are Progressive Networks (Rusu et al., 2016), Learning Without Forgetting (LWF) (Li & Hoiem, 2016) and PathNet (Fernando et al., 2017).

The fourth and more recent method is *generative replay* (Shin et al., 2017; Venkatesan et al., 2017), where a generative model is used to produce samples from previous tasks. This approach has also been referred to as *pseudo-rehearsal*.

## 2.2 CONTINUAL LEARNING FOR GENERATIVE MODELS

Discriminative and generative models do not share the same learning objective and architecture. For this reason, CL strategies for discriminative models are usually not directly applicable to generative models. Continual Learning in the context of generative models remains largely unexplored compared to CL for discriminative models.

Among notable previous work, Seff et al. (2017) successfully apply EWC on the generator of Conditional-GANs (CGANS), after observing that applying the same regularization scheme to a classic GAN leads to catastrophic forgetting. However, their work is based on a scenario where two classes are presented first, and then unique classes come sequentially, e.g the first task is composed of 0 and 1 digits of MNIST dataset, and then is presented with only one digit at a time on the following tasks. This is likely due to the failure of CGANs on single digits, which we observe in our experiments. Moreover, the method is shown to work on CGANs only. Another method for generative Continual Learning is Variational Continual Learning (VCL) (Nguyen et al., 2017), which adapts variational inference to a continual setting. They exploit the online update from one task to another inspired from Bayes' rule. They successfully experiment with VAEs on a single-task scenario. While VCL has the advantage of being a parameter-free method. However, they experiment only on VAEs. Plus, since they use a multi-head architecture, they use specific weights for each task, which need task index for inference. A second method experimented on VAEs is to use a student-teacher method where the student learns the current task while the teacher retains knowledge (Ramapuram et al., 2017). Finally, VASE (Achille et al., 2018) is a third method, also experimented only on VAEs, which allocates spare representational capacity to new knowledge, while protecting previously learned representations from catastrophic forgetting by using snapshots (i.e. weights) of previous model.

A different approach, introduced by Shin et al. (2017) is an adaptation of the *generative replay* method mentioned in Section 2.1. It is applicable to both adversarial and variational frameworks. It uses two generative models: one which acts as a memory, capable of generating all past tasks, and one that learns to generate data from all past tasks and the current task. It has mainly been used as a method for Continual Learning of discriminative models (Shin et al., 2017; Venkatesan et al., 2017; Shah et al., 2018). Recently, Wu et al. (2018a) have developed a similar approach called Memory Replay GANs, where they use Generative Replay combined to replay alignment, a distillation scheme that transfers previous knowledge from a conditional generator to the current one. However they note that this method leads to mode collapse because it could favor learning to generate few class instances rather than a wider range of class instances.

## 3 APPROACH

Typical previous work on Continual Learning for generative models focus on presenting a novel CL technique and comparing it to previous approaches, on one type of generative model (e.g. GAN or VAE). On the contrary, we focus on searching for the best generative model and CL strategy association. For now, empirical evaluation remain the only way to find the best performing combinations. Hence, we compare several existing CL strategies on a wide variety of generative models with the objective of finding the most suited generative model for Continual Learning.

In this process, evaluation metrics are crucial. CL approaches are usually evaluated by computing a metric at the end of each task. Whichever method that is able to maintain the highest performance is best. In the discriminative setting, classification accuracy is the most commonly used metric. Here, as we focus on generative models, there is no consensus on which metric should be used. Thus, we use and compare two quantitative metrics.

The Fréchet Inception Distance (FID) (Heusel et al., 2017) is a commonly used metric for evaluating generative models. It is designed to improve on the Inception Score (IS) (Salimans et al., 2016) which has many intrinsic shortcomings, as well as additional problems when used on a dataset different than ImageNet (Barratt & Sharma, 2018). FID circumvent these issues by comparing the

statistics of generated samples to real samples, instead of evaluating generated samples directly. Heusel et al. (2017) propose using the Fréchet distance between two multivariate Gaussians:

$$FID = \|\mu_r - \mu_g\|^2 + Tr(\Sigma_r + \Sigma_g - 2(\Sigma_r\Sigma_g)^{1/2}), \tag{1}$$

where the statistics $(\mu_r, \Sigma_r)$ and $(\mu_g, \Sigma_g)$ are the activations of a specific layer of a discriminative neural network trained on ImageNet, for real and generated samples respectively. A lower FID correspond to more similar real and generated samples as measured by the distance between their activation distributions. Originally the activation should be taken from a given layer of a given Inception-v3 instance, however this setting can be adapted with another classifier in order to compare a set of models with each other (Li et al., 2017; Lesort et al., 2018).

A different approach is to use labeled generated samples from a generator $G$ (GAN or VAE) to train a classifier and evaluate it afterwards on real data (Lesort et al., 2018). This evaluation, called Fitting Capacity of $G$, is the test accuracy of a classifier trained with $G$'s samples. It measures the generator's ability to train a classifier that generalize well on a testing set, i.e the generator's ability to fit the distribution of the testing set. This method aims at evaluating generative models on complex characteristics of data and not only on their features distribution. In the original paper, the authors annotated samples by generating them conditionally, either with a conditional model or by using one unconditional model for each class. In this paper, we also use an adaptation of the Fitting Capacity where data from unconditional models are labelled by an expert network trained on the dataset.

We believe that using these two metrics is complementary. FID is a commonly used metric based solely on the distribution of images features. In order to have a complementary evaluation, we use the Fitting Capacity, which evaluate samples on a classification criterion rather than features distribution.

For all the progress made in quantitative metrics for evaluating generative models (Borji, 2018), qualitative evaluation remains a widely used and informative method. While visualizing samples provides a instantaneous detection of failure, it does not provide a way to compare two well-performing models. It is not a rigorous evaluation and it may be misleading when evaluating sample variability.

## 4 EXPERIMENTAL SETUP

We now describe our experimental setup: data, tasks, and evaluated approaches. Our code is available online [2].

### 4.1 DATASETS, TASKS, METRICS AND MODELS

Our main experiments use 10 sequential tasks created using the MNIST, Fashion MNIST and CIFAR10 dataset. For each dataset, we define 10 sequential tasks, one task corresponds to learning to generate a new class and all the previous ones (See Fig. 1 for an example on MNIST). Both evaluations, FID and Fitting Capacity of generative models, are computed at the end of each task.

We use 6 different generative models. We experiment with the original and conditional version of GANs (Goodfellow et al., 2014) and VAEs (Kingma & Welling, 2013). We also added WGAN (Arjovsky et al., 2017) and a variant of it WGAN-GP (Gulrajani et al., 2017), as they are commonly used baselines that supposedly improve upon the original GAN.

### 4.2 STRATEGIES FOR CONTINUAL LEARNING

We focus on strategies that are usable in both the variational and adversarial frameworks. We use 3 different strategies for Continual Learning of generative models, that we compare to 3 baselines. Our experiments are done on 8 seeds with 50 epochs per tasks for MNIST and Fashion MNIST

---

[2]https://github.com/anonymous-authors-2018/Generative_models_from_the_perspective_of_Continual_learning

using Adam (Kingma & Ba, 2014) for optimization (for hyper-parameter settings, see Appendix F). For CIFAR10, we experimented with the best performing CL strategy.

The first baseline is Fine-tuning, which consists in ignoring catastrophic forgetting and is essentially a lower bound of the performance. Our other baselines are two upper bounds: Upperbound Data, for which one generative model is trained on joint data from all past tasks, and Upperbound Model, for which one separate generator is trained for each task.

For Continual Learning strategies, we first use a vanilla Rehearsal method, where we keep a fixed number of samples of each observed task, and add those samples to the training set of the current generative model. We balance the resulting dataset by copying the saved samples so that each class has the same number of samples. The number of samples selected, here 10, is motivated by the results in Fig. 7a and 7b, where we show that 10 samples per class is enough to get a satisfactory but not maximal validation accuracy for a classification task on MNIST and Fashion MNIST. As the Fitting Capacity share the same test set, we can compare the original accuracy with 10 samples per task to the final fitting capacity. A higher Fitting capacity show that the memory prevents catastrophic forgetting. Equal Fitting Capacity means overfitting of the saved samples and lower Fitting Capacity means that the generator failed to even memorize these samples.

We also experiment with EWC. We followed the method described by Seff et al. (2017) for GANs, i.e. the penalty is applied only on the generator's weights , and for VAEs we apply the penalty on both the encoder and decoder. As tasks are sequentially presented, we choose to update the diagonal of the Fisher information matrix by cumulatively adding the new one to the previous one. The last method is Generative Replay, described in Section 2.2. Generative replay is a dual-model approach where a "frozen" generative model $G_{t-1}$ is used to sample from previously learned distributions and a "current" generative model $G_t$ is used to learn the current distribution and $G_{t-1}$ distribution. When a task is over, the $G_{t-1}$ is replaced by a copy of $G_t$ , and learning can continue.

## 5 RESULTS

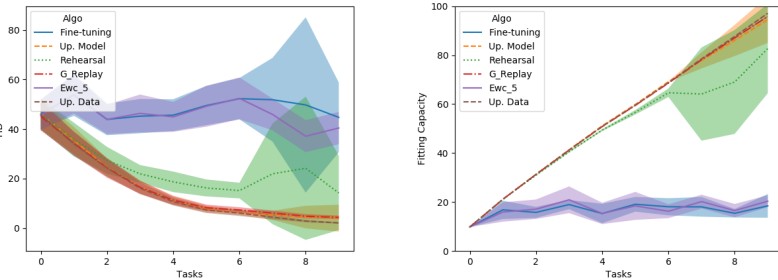

Figure 2: Comparison, averaged over 8 seeds, between FID results(left, lower is better) and Fitting Capacity results (right, higher is better) with GAN trained on MNIST.

The figures we report show the evolution of the metrics through tasks. Both FID and Fitting Capacity are computed on the test set. A well performing model should increase its Fitting Capacity and decrease its FID. We observe a strong correlation between the Fitting Capacity and FID (see Fig. 2 for an example on GAN for MNIST and Appendix C for full results). Nevertheless, Fitting Capacity results are more stable: over the 8 random seeds we used, the standard deviations are less important than in the FID results. For that reason, we focus our interpretation on the Fitting Capacity results.

### 5.1 MNIST AND FASHION MNIST RESULTS

#### 5.1.1 MAIN RESULTS

Our main results with Fitting Capacity are displayed in Fig. 3 and Table 1. The best combination was Generative Replay + GAN with a mean Fitting Capacity of $95.81\%$ on MNIST and $81.52\%$ on Fashion MNIST. The relative performance of each CL method on GAN can be analyzed class

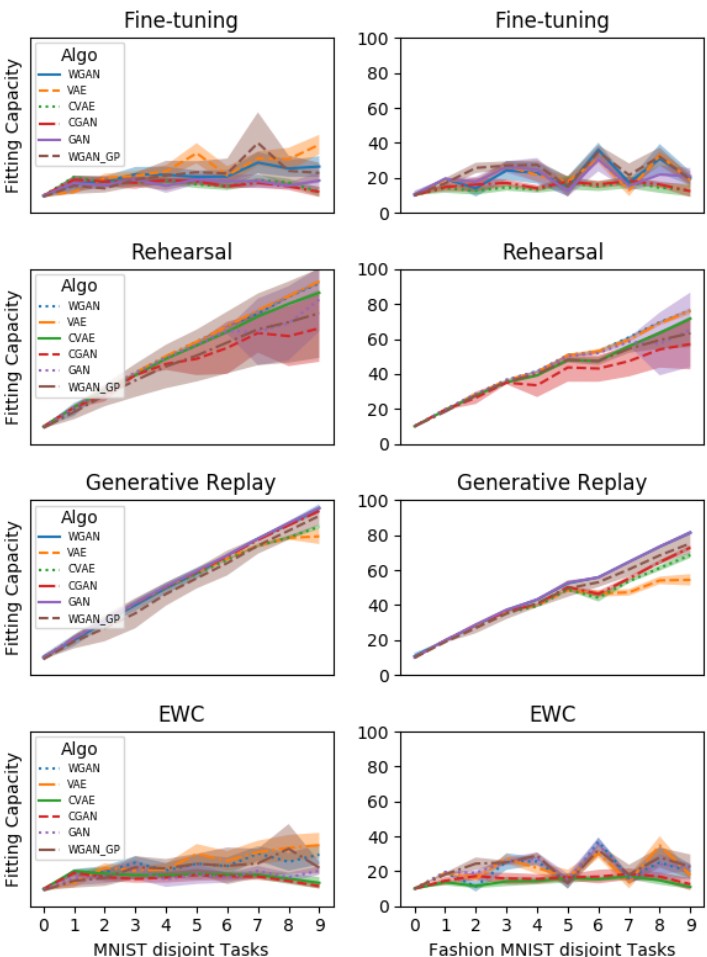

Figure 3: Means and standard deviations over 8 seeds of Fitting Capacity metric evaluation of VAE, CVAE, GAN, CGAN and WGAN. The four considered CL strategies are: Fine Tuning, Generative Replay, Rehearsal and EWC. The setting is 10 disjoint tasks on MNIST and Fashion MNIST.

by class in Fig. 4. We observe that, for the adversarial framework, Generative Replay outperforms other approaches by a significant margin. However, for the variational framework, the Rehearsal approach was the best performing. The Rehearsal approach worked quite well but is unsatisfactory for CGAN and WGAN-GP. Indeed, the Fitting Capacity is lower than the accuracy of a classifier trained on 10 samples per classes (see Fig. 7a and 7b in Appendix). In our setting, EWC is not able to overcome catastrophic forgetting and performs as well as the naive Fine-tuning baseline which is contradictory with the results of Seff et al. (2017) who found EWC successful in a slightly different setting. We replicated their result in a setting where there are two classes by tasks (see Appendix E for details), showing the strong effect of task definition.

In Seff et al. (2017) authors already found that EWC did not work with non-conditional models but showed successful results with conditional models (i.e. CGANs). The difference come from the experimental setting. In Seff et al. (2017), the training sequence start by a task with two classes. Hence, when CGAN is trained it is possible for the Fisher Matrix to understand the influence of the class-index input vector $c$. In our setting, since there is only one class at the first task, the Fisher matrix can not get the importance of the class-index input vector $c$. Hence, as for non conditional models, the Fisher Matrix is not able to protect weights appropriately and at the end of the second task the model has forgot the first task. Moreover, since the generator forgot what it learned at the first task, it is only capable of generating samples of only one class. Then, the Fisher Matrix will still not get the influence of $c$ until the end of the sequence. Moreover, we show that even by

Table 1: Mean and standard deviations for Fitting Capacity (in %) metric evaluation on last task of 10 disjoint task setting, on MNIST and Fashion MNIST, over 8 seeds.

| Strategy | Dataset | GAN | CGAN | WGAN | WGAN-GP | VAE | CVAE |
|---|---|---|---|---|---|---|---|
| Fine-tuning | MNIST | $18.43_{\pm4.85}$ | $11.93_{\pm2.97}$ | $23.17_{\pm5.66}$ | $22.79_{\pm5.75}$ | $38.98_{\pm5.57}$ | $11.96_{\pm2.56}$ |
| EWC | - | $20.34_{\pm2.39}$ | $11.53_{\pm1.42}$ | $29.57_{\pm5.59}$ | $22.00_{\pm3.39}$ | $34.93_{\pm7.06}$ | $13.37_{\pm3.28}$ |
| Rehearsal | - | $82.69_{\pm18.21}$ | $66.14_{\pm19.2}$ | $92.05_{\pm0.64}$ | $74.79_{\pm25.25}$ | $92.99_{\pm0.64}$ | $86.47_{\pm1.69}$ |
| Generative Replay | - | $\mathbf{95.81}_{\pm0.31}$ | $93.89_{\pm0.35}$ | $95.41_{\pm2.41}$ | $91.12_{\pm5.09}$ | $79.38_{\pm4.40}$ | $84.95_{\pm1.24}$ |
| Upperbound Model | - | $94.50_{\pm9.51}$ | $96.84_{\pm3.22}$ | $95.72_{\pm6.93}$ | $79.41_{\pm27.85}$ | $97.82_{\pm0.17}$ | $97.89_{\pm0.12}$ |
| Upperbound Data | - | $97.10_{\pm0.13}$ | $96.65_{\pm0.21}$ | $96.76_{\pm0.29}$ | $84.79_{\pm27.76}$ | $96.88_{\pm0.27}$ | $96.17_{\pm0.19}$ |
| Fine-tuning | Fashion MNIST | $20.82_{\pm4.69}$ | $12.30_{\pm3.33}$ | $19.68_{\pm3.92}$ | $18.75_{\pm2.58}$ | $18.60_{\pm4.24}$ | $12.82_{\pm3.55}$ |
| EWC | - | $22.22_{\pm2.03}$ | $12.58_{\pm3.48}$ | $19.81_{\pm4.18}$ | $22.63_{\pm6.91}$ | $17.70_{\pm1.83}$ | $11.00_{\pm1.16}$ |
| Rehearsal | - | $65.34_{\pm21.3}$ | $57.12_{\pm14.4}$ | $76.32_{\pm0.33}$ | $63.28_{\pm7.9}$ | $76.03_{\pm1.77}$ | $71.73_{\pm1.29}$ |
| Generative Replay | - | $\mathbf{81.52}_{\pm0.87}$ | $72.98_{\pm1.22}$ | $81.50_{\pm1.26}$ | $75.37_{\pm5.49}$ | $54.49_{\pm3.24}$ | $68.70_{\pm1.71}$ |
| Upperbound Model | - | $77.93_{\pm15.07}$ | $80.96_{\pm0.69}$ | $73.20_{\pm5.63}$ | $65.5_{\pm2.69}$ | $78.64_{\pm1.36}$ | $79.15_{\pm0.96}$ |
| Upperbound Data | - | $83.27_{\pm0.41}$ | $80.09_{\pm0.94}$ | $83.29_{\pm0.52}$ | $81.5_{\pm0.50}$ | $80.21_{\pm0.79}$ | $79.51_{\pm0.55}$ |

starting with 2 classes, when there is only one class for the second task, the Fisher matrix is not able to protect the class from the second task in the third task. (see Figure 11).

Our results do not give a clear distinction between conditional and unconditional models. However, adversarial methods perform significantly better than variational methods. GANs variants are able to produce better, sharper quality and variety of samples, as observed in Fig. 14 and 15 in Appendix G. Hence, adversarial methods seem more viable for CL. We can link the accuracy from 7a and 7b to the Fitting Capacity results. As an example, we can estimate that GAN with Generative Replay is equivalent for both datasets to a memory of approximately 100 samples per class.

### 5.1.2 COROLLARY RESULTS

Catastrophic forgetting can be visualized in Fig.4. Each square's column represent the task index and each row the class, the color indicate the Fitting Capacity (FC). Yellow squares show a high FC, blue one show a low FC. We can visualize both the performance of VAE and GAN but also the performance evolution for each class. For Generative Replay, at the end of the task sequence, VAE decreases its performance in several classes when GAN does not. For Rehearsal it is the opposite. Concerning the high performance of original GAN and WGAN with Generative Replay, they benefit from their samples quality and their stability. In comparison, samples from CGAN and WGAN-GP are more noisy and samples from VAE and CVAE more blurry (see in appendix 14). However in the Rehearsal approach GANs based models seems much less stable (See Table 1 and Figure 3). In this setting the discriminative task is almost trivial for the discriminator which make training harder for the generator. In opposition, VAE based models are particularly effective and stable in the Rehearsal setting (See Fig. 4b). Indeed, their learning objective (pixel-wise error) is not disturbed by a low samples variability and their probabilistic hidden variables make them less prone to overfit.

However the Fitting Capacity of Fine-tuning and EWC in Table 1 is higher than expected for unconditional models. As the generator is only able to produce samples from the last task, the Fitting capacity should be near $10\%$. This is a downside of using an expert for annotation before computing the Fitting Capacity. Fuzzy samples can be wrongly annotated, which can artificially increase the labels variability and thus the Fitting Capacity of low performing models, e.g., VAE with Fine-tuning. However, this results stay lower than the Fitting Capacity of well performing models.

Incidentally, an important side result is that the Fitting capacity of conditional generative models is comparable to results of Continual Learning classification. Our best performance in this setting is with CGAN: $94.7\%$ on MNIST and $75.44\%$ on Fashion MNIST . In a similar setting with 2 sequential tasks, which is arguably easier than our setting (one with digits from 0,1,2,3,4 and another with 5,6,7,8,9), He & Jaeger (2018) achieve a performance of $94.91\%$. This shows that using generative models for CL could be a competitive tool in a classification scenario. It is worth noting that we did not compare our results of unconditional models Fitting Capacity with classification state of the art. Indeed, in this case, the Fitting capacity is based on an annotation from an expert not trained in a continual setting. The comparison would then not be fair.

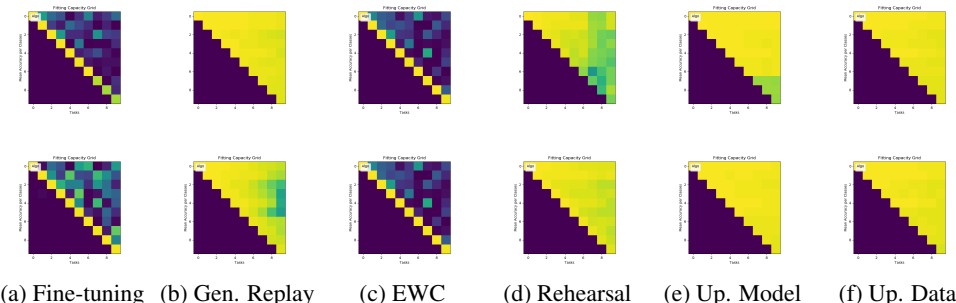

(a) Fine-tuning  (b) Gen. Replay  (c) EWC  (d) Rehearsal  (e) Up. Model  (f) Up. Data

Figure 4: Fitting Capacity results for GAN (top) and VAE (bottom) on MNIST. Each square is the accuracy on one class for one task. Abscissa is the task index (left: 0 , right: 9) and orderly is the class index (top: 0, down: 9). The accuracy is proportional to the color (dark blue : 0%, yellow 100%)

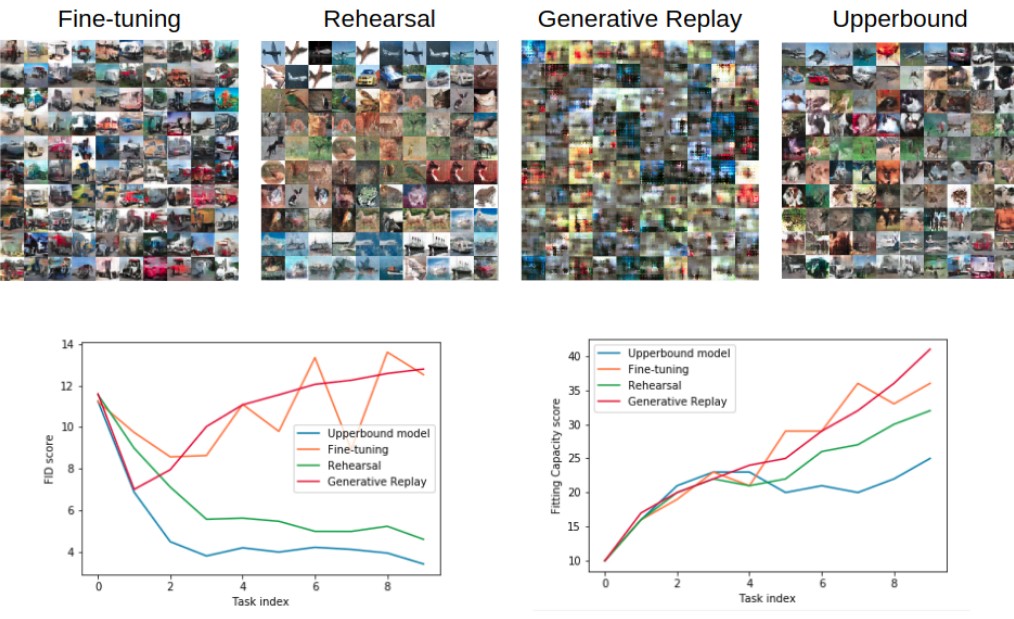

Figure 5: Fitting capacity and FID score of Continual Learning methods applied to WGAN_GP, on CIFAR10. For each method, images sampled after the 10 sequential tasks are displayed.

## 5.2 CIFAR10 RESULTS

In this experiment, we selected the best performing CL methods on MNIST and Fashion MNIST, Generative Replay and Rehearsal, and tested it on the more challenging CIFAR10 dataset. We compared the two method to naive Fine-tuning, and to Upperbound Model (one generator for each class). The setting remains the same, one task for each category, for which the aim is to avoid forgetting of previously seen categories. We selected WGAN-GP because it produced the most satisfying samples on CIFAR10 (see Fig. 16 in Appendix G).

Results are provided in Fig. 5, where we display images sampled after the 10 sequential tasks, and FID + Fitting Capacity curves throughout training. The Fitting Capacity results show that all four methods fail to generate images that allow to learn a classifier that performs well on real CIFAR10 test data. As stated for MNIST and Fashion MNIST, with non-conditional models, when the Fitting Capacity is low, it can been artificially increased by automatic annotation which make the difference between curves not significant in this case. Naive Fine-tuning catastrophically forgets previous tasks, as expected. Rehearsal does not yield satisfactory results. While the FID score shows improvement at each new task, visualization clearly shows that the generator copies samples in memory, and

suffers from mode collapse. This confirms our intuition that Rehearsal overfits to the few samples kept in memory. Generative Replay fails; since the dataset is composed of real-life images, the generation task is much harder to complete. We illustrate its failure mode in Figure 17 in Appendix G. As seen in Task 0, the generator is able to produce images that roughly resemble samples of the category, here planes. As tasks are presented, minor generation errors accumulated and snowballed into the result in task 9: samples are blurry and categories are indistinguishable. As a consequence, the FID improves at the beginning of the training sequence, and then deteriorates at each new task. We also trained the same model separately on each task, and while the result is visually satisfactory, the quantitative metrics show that generation quality is not excellent.

These negative results shows that training a generative model on a sequential task scenario does not reduce to successfully training a generative model on all data or each category, and that state-of-the-art generative models struggle on real-life image datasets like CIFAR10. Designing a CL strategy for these type of datasets remains a challenge.

## 6 DISCUSSION

Besides the quantitative results and visual evaluation of the generated samples, the evaluated strategies have, by design, specific characteristics relevant to CL that we discuss here.

Rehearsal violates the data availability assumption, often required in CL scenarios, by recording part of the samples. Furthermore the risk of overfitting is high when only few samples represent a task, as shown in the CIFAR10 results. EWC and Generative Replay respect this assumption. EWC has the advantage of not requiring any computational overload during training, but this comes at the cost of computing the Fisher information matrix, and storing its values as well as a copy of previous parameters. The memory needed for EWC to save information from the past is twice the size of the model which may be expensive in comparison to rehearsal methods. Nevertheless, with Rehearsal and Generative Replay, the model has more and more samples to learn from at each new task, which makes training more costly.

Another point we discuss is about a recently proposed metric (Wu et al., 2018a) to evaluate CL for generative models. Their evaluation is defined for conditional generative models. For a given label $l$, they sample images from the generator conditioned on $l$ and feed it to a pre-trained classifier. If the predicted label of the classifier matches $l$, then it is considered correct. In our experiment we find that it gives a clear advantage to rehearsal methods. As the generator may overfit the few samples kept in memory, it can maximizes the evaluation proposed by Wu et al. (2018b), while not producing diverse samples. We present this phenomenon with our experiments in appendix D. Nevertheless, even if their metric is unable to detect mode collapse or overfitting, it can efficiently expose catastrophic forgetting in conditional models.

## 7 CONCLUSION AND FUTURE WORK

In this paper, we experimented with the viability and effectiveness of generative models on Continual Learning (CL) settings. We evaluated the considered approaches on commonly used datasets for CL, with two quantitative metrics. Our experiments indicate that on MNIST and Fashion MNIST, the original GAN combined to the Generative Replay method is particularly effective. This method avoids catastrophic forgetting by using the generator as a memory to sample from the previous tasks and hence maintain past knowledge. Furthermore, we shed light on how generative models can learn continually with various methods and present successful combinations. We also reveal that generative models do not perform well enough on CIFAR10 to learn continually. Since generation errors accumulate, they are not usable in a continual setting. The considered approaches have limitations: we rely on a setting where task boundaries are discrete and given by the user. In future work, we plan to investigate automatic detection of tasks boundaries. Another improvement would be to experiment with smoother transitions between tasks, rather than the disjoint tasks setting.

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

## A   SAMPLES AT EACH STEP

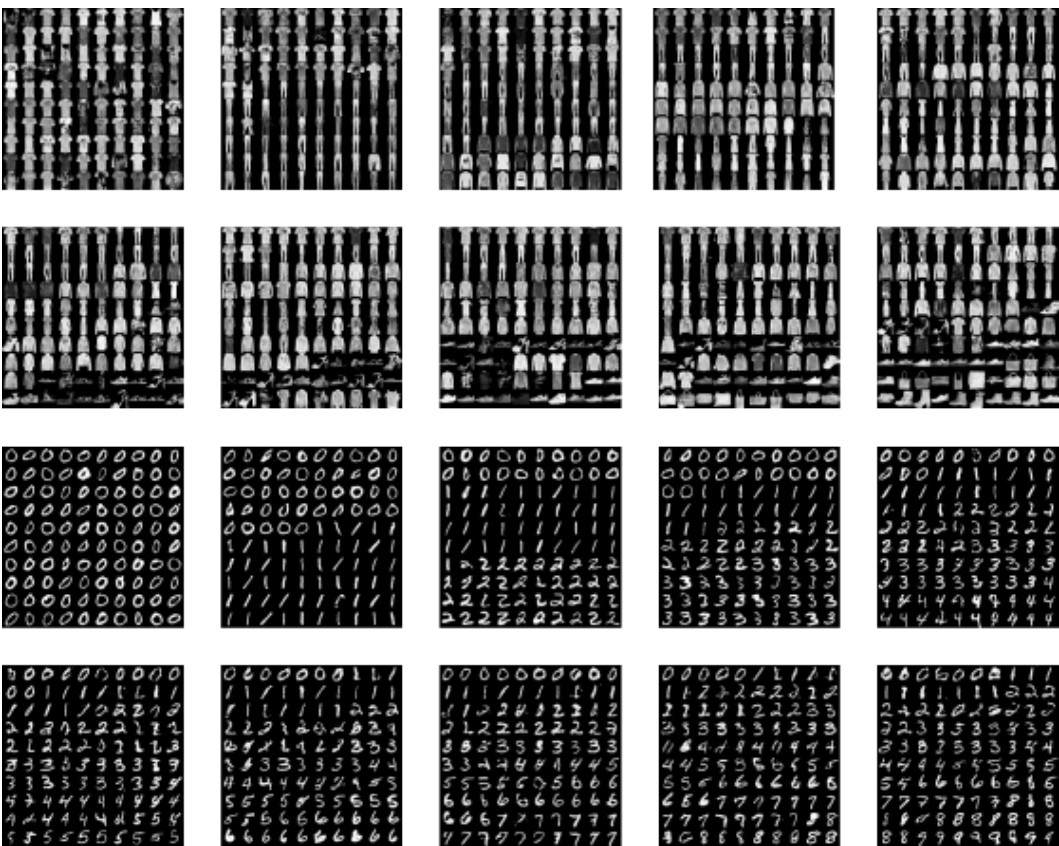

Figure 6: Samples of a well performing solution : GAN + Generative Replay for each step in a sequence of 10 tasks with MNIST and Fashion MNIST.

## B   CLASSIFIERS PERFORMANCES

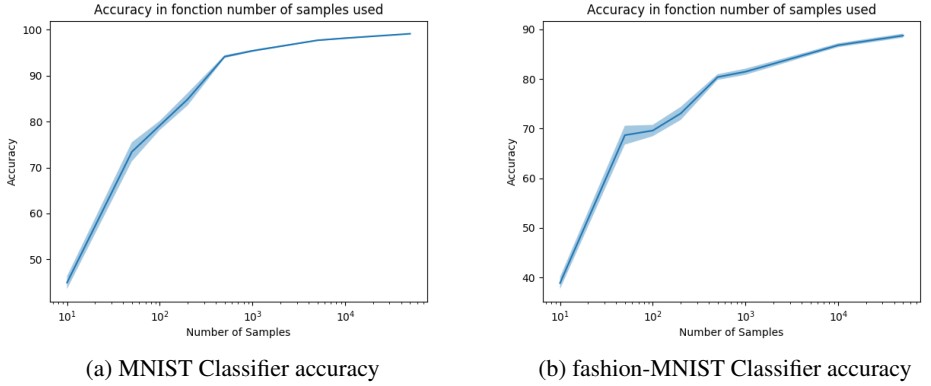

(a) MNIST Classifier accuracy

(b) fashion-MNIST Classifier accuracy

Figure 7: Test set classification accuracy as a function of number of training samples, on MNIST.

Fig. 7a and 7b make possible to estimate the number of samples needed to solve the full task. Furthermore by comparing against the fitting capacity we can estimate how many different images of the dataset a generator can produce.

# C RESULTS PER MODEL

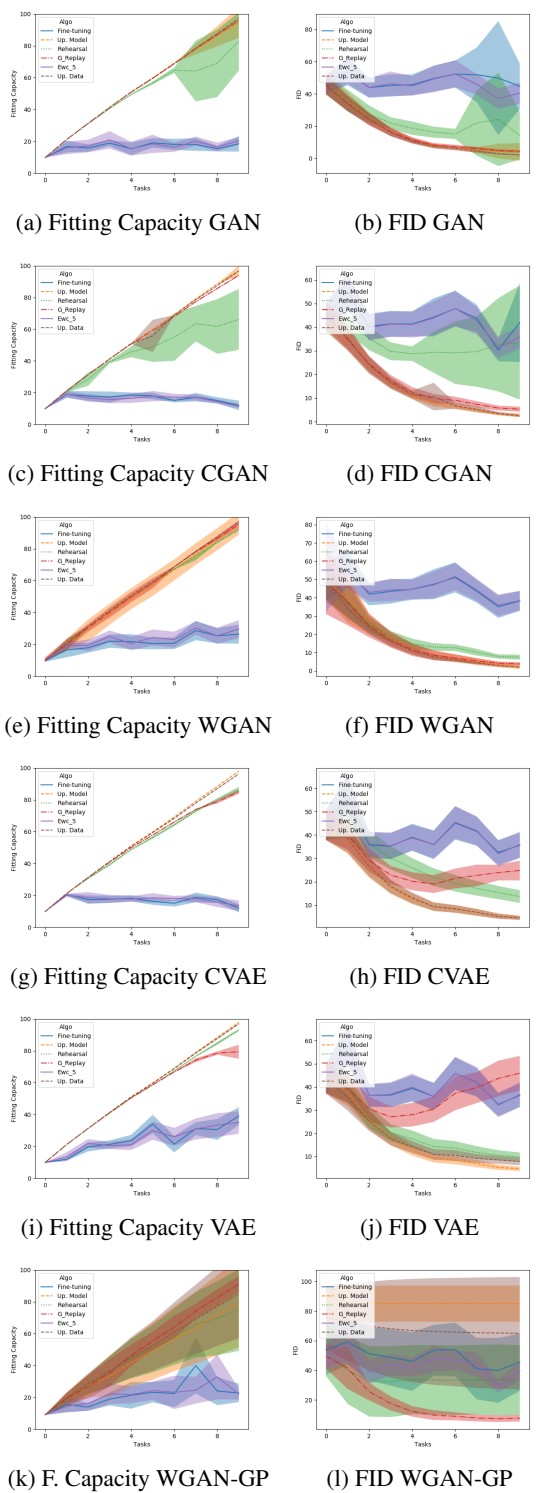

(a) Fitting Capacity GAN

(b) FID GAN

(c) Fitting Capacity CGAN

(d) FID CGAN

(e) Fitting Capacity WGAN

(f) FID WGAN

(g) Fitting Capacity CVAE

(h) FID CVAE

(i) Fitting Capacity VAE

(j) FID VAE

(k) F. Capacity WGAN-GP

(l) FID WGAN-GP

Figure 8: Comparison of the Fitting Capacity and FID results on MNIST.

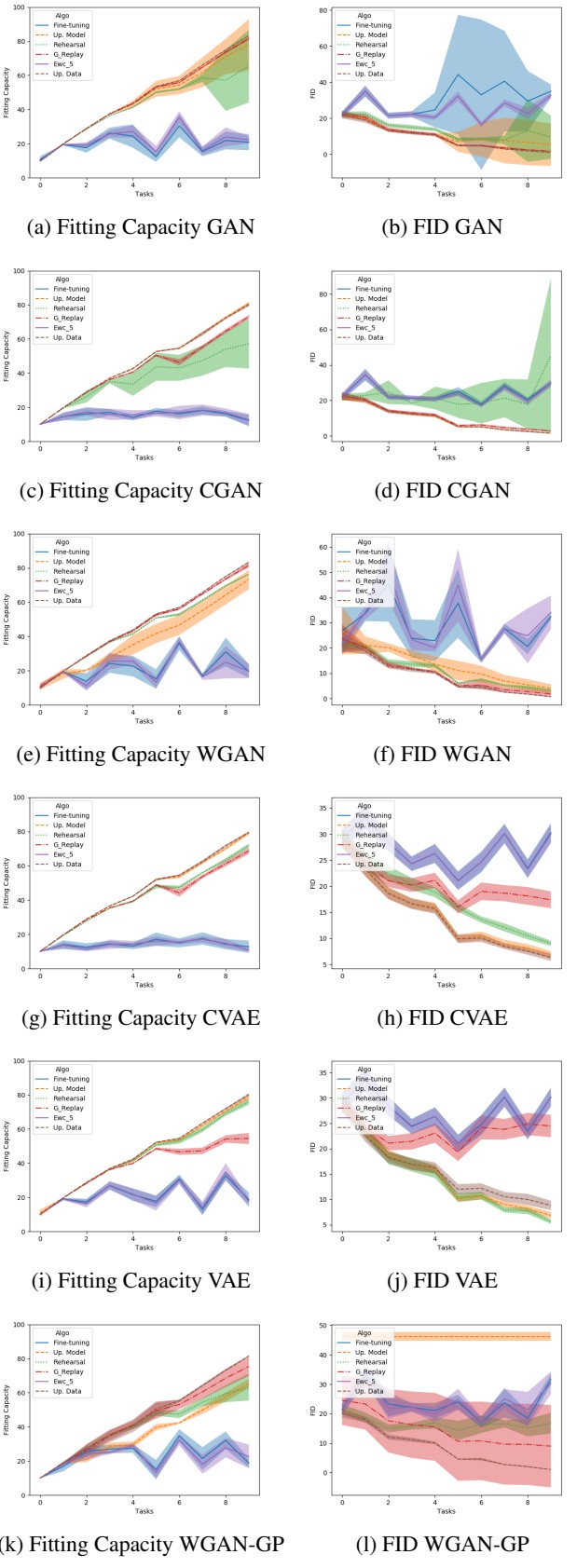

(a) Fitting Capacity GAN

(b) FID GAN

(c) Fitting Capacity CGAN

(d) FID CGAN

(e) Fitting Capacity WGAN

(f) FID WGAN

(g) Fitting Capacity CVAE

(h) FID CVAE

(i) Fitting Capacity VAE

(j) FID VAE

(k) Fitting Capacity WGAN-GP

(l) FID WGAN-GP

Figure 9: Comparison of the Fitting Capacity and FID results on Fashion MNIST.

## D COMPARISON WITH (WU ET AL., 2018A)

Table 2: Our results using the metric proposed by Wu et al. (2018a). Rehearsal, even thought suffers from mode collapse, performs as good as Generative Replay, which visually produce better samples.

| Strategy | Dataset | CVAE | CGAN |
|----------|---------|--------|--------|
| Rehearsal | Mnist | 99.86% | 95.72% |
| Generative Replay | - | 99.70% | 99.26% |
| Ewc | - | 10.78% | 10.54% |
| Baseline | - | 10.70% | 10.52% |
| Rehearsal | Fashion | 94.42% | 92.36% |
| Generative Replay | - | 88.64% | 89.98% |
| Ewc | - | 10.62% | 10.50% |
| Baseline | - | 10.68% | 10.60% |

## E REPRODUCTION OF RESULTS IN (SEFF ET AL., 2017)

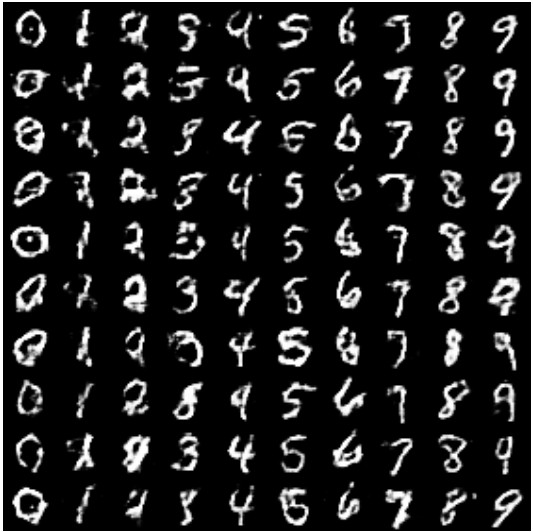

Figure 10: CGAN augmented with EWC. MNIST samples after 5 sequential tasks of 2 digits each. Catastrophic forgetting in avoided.

## F HYPERPARAMETERS

Table 3: Hyperparameters for MNIST and Fashion MNIST all models ( all CL strategies have the same training hyper parameters)

| Model Datasets | Epochs | Lr | n_critic | beta1 | beta2 | Batch | lambda | clipping value |
|----------------|--------|------|----------|-------|-------|-------|--------|----------------|
| GAN | 50 | 2e-4 | 1 | 5e-1 | 0.999 | 64 | - | - |
| CGAN | 50 | 2e-4 | 1 | 5e-1 | 0.999 | 64 | - | - |
| VAE | 50 | 2e-4 | 1 | 5e-1 | 0.999 | 64 | - | - |
| CVAE | 50 | 2e-4 | 1 | 5e-1 | 0.999 | 64 | - | - |
| WGAN | 50 | 2e-4 | 2 | 5e-1 | 0.999 | 64 | - | 0.01 |
| WGAN_GP | 50 | 2e-4 | 2 | 5e-1 | 0.999 | 64 | 0.25 | - |
| Classifier | 50 | 0.5 | - | 5e-1 | 0.999 | 64 | - | - |

## G SAMPLES

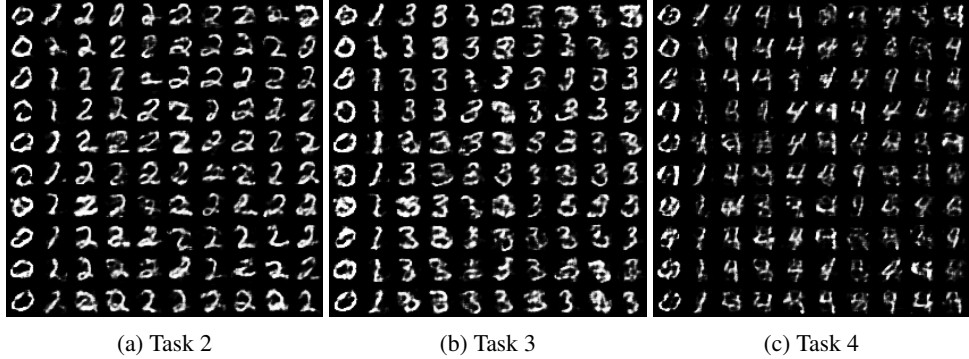

(a) Task 2                    (b) Task 3                    (c) Task 4

Figure 11: Reproduction of EWC experiment (Seff et al., 2017) with four tasks. First task with 0 and 1 digits, then digits of 2 for task 2, digits of 3 for task 3 etc. When task contains only one class, the Fisher information matrix cannot capture the importance of the class-index input vector because it is always fixed to one class. This problem makes the learning setting similar to a non-conditional models one which is known to not work (Seff et al., 2017). As a consequence 0 and 1 are well protected when following classes are not.

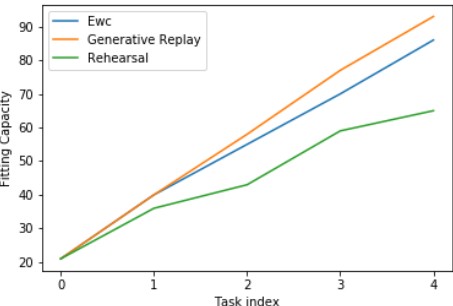

Figure 12: CGAN results with EWC, Rehearsal and Generative Replay, on 5 sequential tasks of 2 digits each. EWC performs well, compared to the results obtained on a 10 sequential task setting.

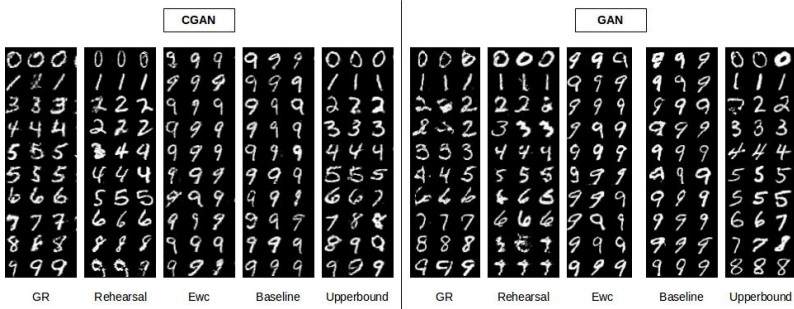

Figure 13: Samples from GAN and Conditional-GAN for each Continual Learning strategy. Upperbound refers to Upperbound Model.

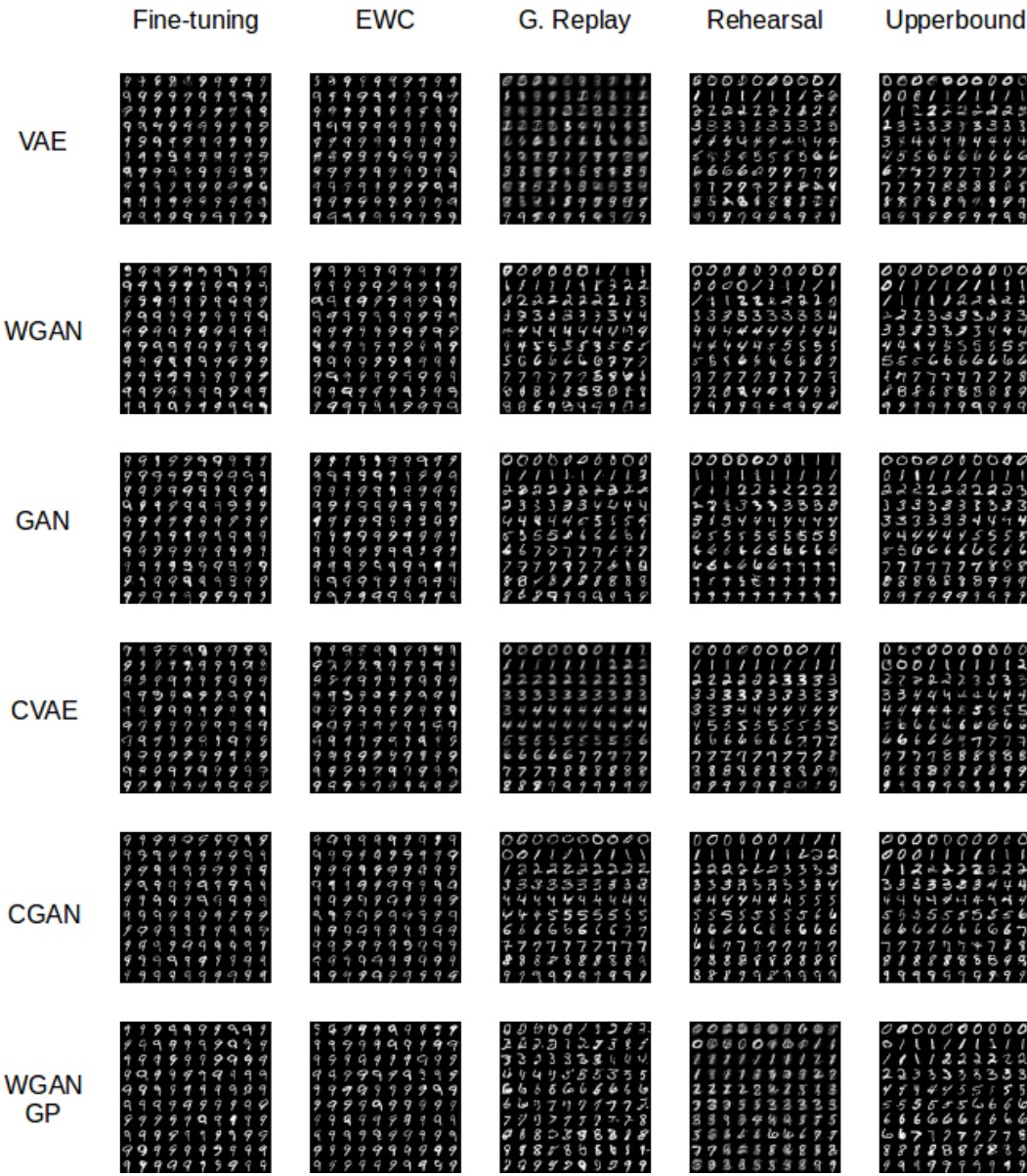

Figure 14: MNIST samples for each generative model and each Continual Learning strategy, at the end of training on 10 sequential tasks. The goal is to produce samples from all categories.

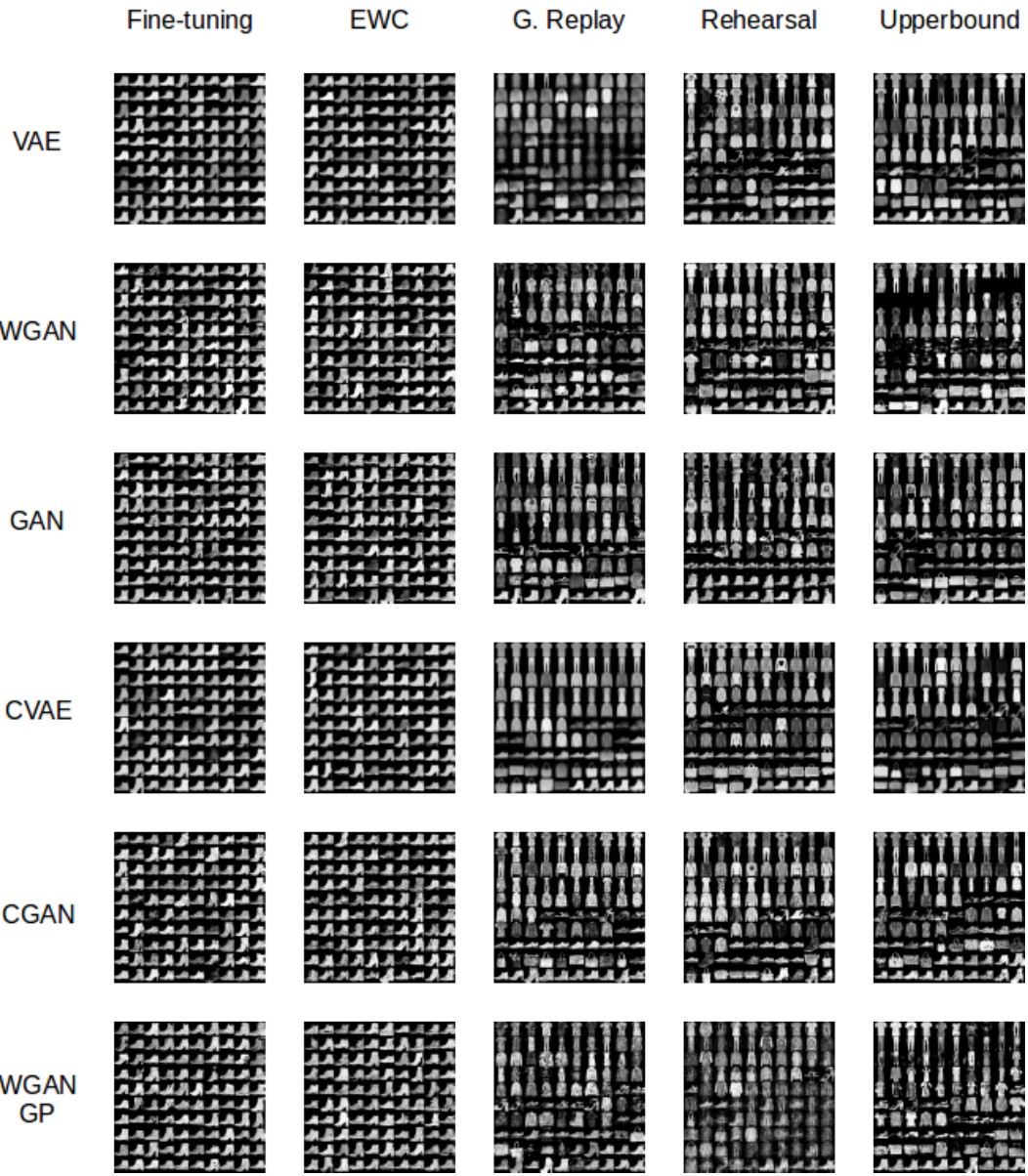

Figure 15: Fashion MNIST samples for each generative model and each Continual Learning strategy, at the end of training on 10 sequential tasks. The goal is to produce samples from all categories.

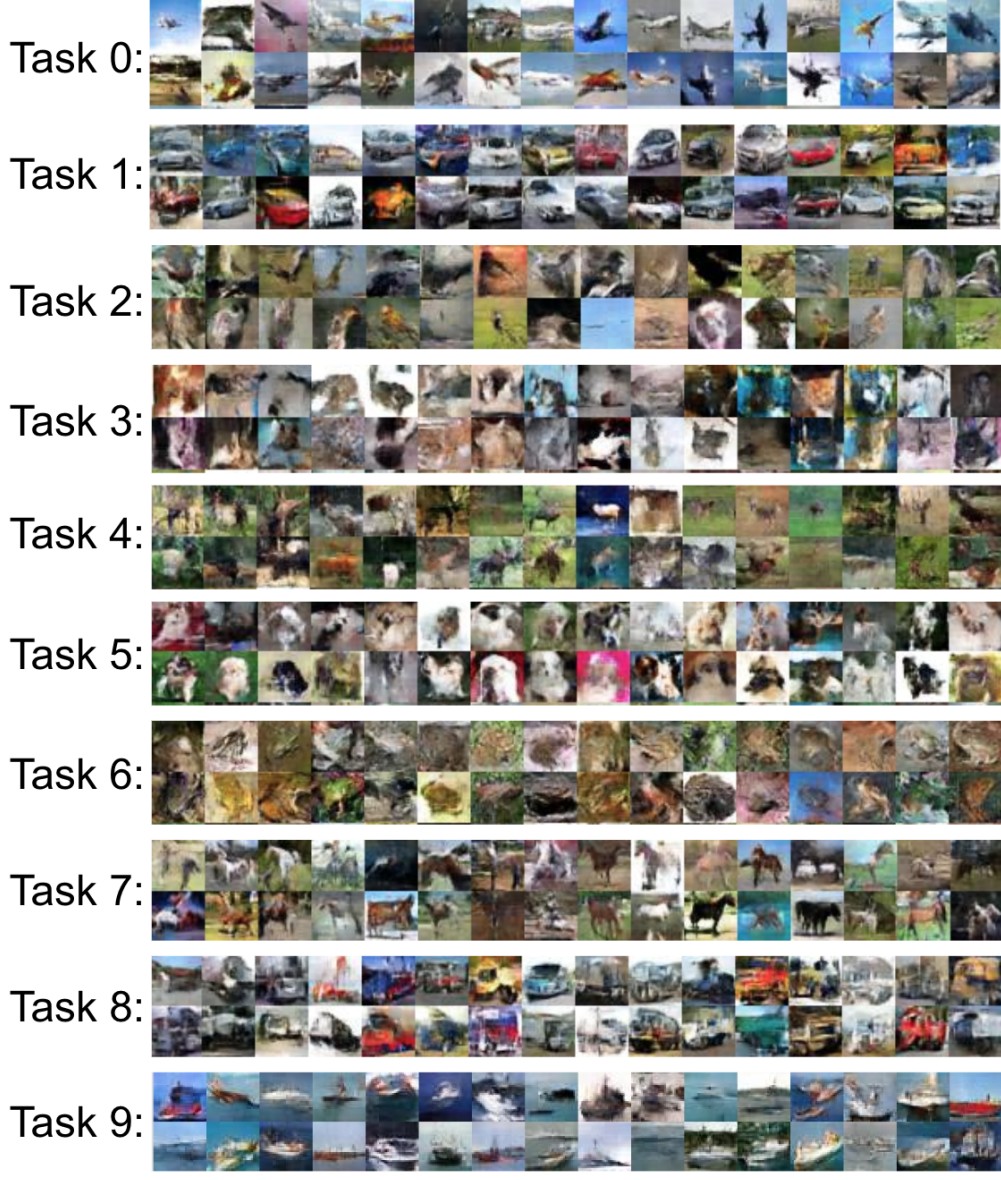

Figure 16: WGAN-GP samples on CIFAR10, with on training for each separate category. The implementation we used is available here: `https://github.com/caogang/wgan-gp`. Classes, from 0 to 9, are planes, cars, birds, cats, deers, dogs, frogs, horses, ships and trucks.

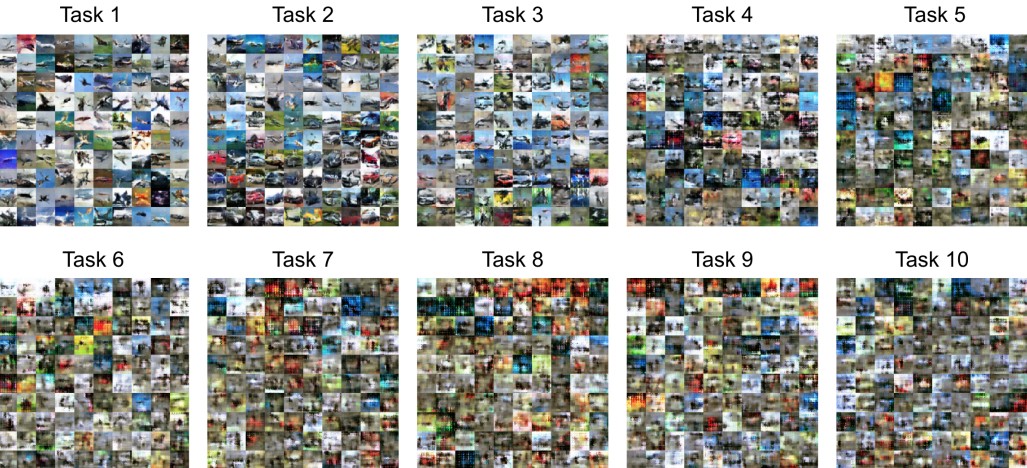

Figure 17: WGAN-GP samples on 10 sequential tasks on CIFAR10, with Generative Replay. Classes, from 0 to 9, are planes, cars, birds, cats, deers, dogs, frogs, horses, ships and trucks. We observe that generation errors snowballs as tasks are encountered, so that the images sampled after the last task are completely blurry.

