# OpenReview forum: "Generative Models from the perspective of Continual Learning"
_ICLR.cc/2019/Conference_

### Official Review · AnonReviewer2 · 2018-11-01
**Potentially nice empirical study, but needs more work on experimental analysis and discussion**

**Rating:** 4
**Confidence:** 4

**Review:**

This paper presents an empirical evaluation of continual learning approaches for generative modelling. Noting that much of previous work focuses on supervised tasks, the paper evaluates various combinations of continual learning strategies (EWC, rehearsal/replay-based, or generative replay) and generative models (GANs or likelihood-based).
The experiments evaluate all combinations on MNIST and Fashion MNIST, and the resulting best-performing combination on CIFAR.
The paper is well-written and structured, and although there are no new proposed algorithms or measures, I think this has the potential to be a useful empirical study on the relatively unstudied topic of continual learning with generative models.

However, my main concern is in the detail of analysis and discussion: for an empirical study, it would be much more beneficial to empirically investigate *why* certain combinations are more effective than others. For example:
- Is the reason GANs are better than likelihood models with generative replay purely because of sample quality? Or is it sufficient for the generator to learn some key characteristics for a class that lead to sufficient discriminability?
- Why is rehearsal better for likelihood models? (And how does this relate to the hypothesis of overfitting to a few real examples?)

The CIFAR-10 results also require more work - it is unclear why the existing approaches could not be made to work, and whether this is a fundamental deficiency in the existing approaches or other factors (hyperparameters, architecture choices, lack of time, etc). Presuming the sample quality is as good as in the WGAN-GP work (given the original implementation is used for experiments), why is this insufficient for generative replay? More detailed analysis / discussion, or another combinatorial study, would help for CIFAR too.

Some comments:
- The poor performance of EWC across the board is concerning. Was this implemented by computing the Fisher of the ELBO with respect to parameters? Was the empirical or true Fisher used? Why does the performance appear so poor compared to Seff et al (2017)? This suggests that either more thought is required on how to best protect parameters of generative models, or the baseline has not been properly implemented/tuned.
- Given this is an entirely empirical study, I would strongly encourage the authors to release code sooner than the acceptance deadline - this can be achieved using an anonymous repository.
- Figure 2 and 3 plots are a little difficult to parse without axis labels.

---

> ### Author Response · Authors · 2018-11-26
> **Answer**
>
> Dear Reviewer,
> Thanks for the thorough review, the comments are particularly helpful.
> - "Is the reason GANs are better than likelihood models with generative replay purely because of sample quality? Or is it sufficient for the generator to learn some key characteristics for a class that leads to sufficient discriminability?"
> The samples in appendix give an insight for this (even if it might be a bit small). The samples from VAEs based models are more blurry than samples from GANs based models. The advantage of GANs is mainly due to the quality and variability of samples. We don't think that it only comes from learned discriminable features since models as GANs (a fortiori for non-conditional models) have no information about the feature that discriminates one class to another only about feature that discriminate training images from generated one.
> We added these comments in the paper.
> - "Why is rehearsal better for likelihood models? (And how does this relate to the hypothesis of overfitting to a few real examples?)"
> The problem of VAE is that their samples are not as sharp as their training data which produce some accumulated approximation when we learn from generated samples recursively (which explain that they are less performant than GAN based models). However, VAE has the strength that, thanks to their probabilistic hidden variable, they are less prone to overfitting. This quality combined to learning on real samples stored by rehearsal method make them particularly effective in a Rehearsal setting.
> We added these comments in the paper.
> - Concerning CIFAR10 : You can refer to Figure 16 to see that our model was generating acceptable images but not good enough for Generative Replay. Plus, we added the combination WGAN\_GP + Rehearsal and WGAN\_GP + Fine-tuning. This additional experiment confirms the overfitting intuition suggested by Rehearsal results on MNIST and Fashion MNIST: see Figure 5 where Rehearsal samples clearly show mode collapse on the few samples kept in memory.
> - About the concerns regarding EWC :
> We compute the empirical Fisher information matrix: we approximate the expected value of the square of the score function by the empirical mean over all samples. For GANs, the score function is the part of the loss function that involves the generator, as we follow the approach in Seff et al. For VAEs, we use the log of the loss function (which is the indeed the ELBO).
> Seff et al (2017) already discuss the fact that EWC does not work with non-conditional models (the argument applies for GAN and VAE), so the poor performance of EWC with GAN, WGAN, and VAE is not surprising.
> For C-GAN, we added in appendix a similar experiment to Seff et al (2017) with 5 tasks with 2 classes each time, which worked well (catastrophic forgetting avoided). We added this experiment to show that our implementation of EWC is correct. We added results with 2 classes at the first task and after one task of one class to reproduce it more rigorously, see Figure 11. Note that we already extensively tried to tune the lambda hyperparameter of EWC but we were not successful in finding a configuration that worked better than reported results on the disjoint setting.
> We think that the fact that EWC does not work in our setting comes from the computation of the Fisher matrix.
> The input of the C-GAN is random noise z, and the class-conditional input c. At no instant, in our setting, it is possible for the Fisher matrix to get information about the link between the class-conditional input c and the output image (and thus its class).
> In fact, at task 0, no matter the class-conditional input c, the generator will produce the same kind of output (images of the first class). Consequently, the fisher matrix will not be able to protect class-conditional weights correctly which will lead the generator, at task 1, to forget what it learned previously and it will then only being able to generate samples of second-class whatever the value of c. The same argument applies recursively for the following tasks.
> Note that in Wu et al. (2018), the reported results of C-GAN are also very poor compared to Seff et al. The setting in Wu et al. is the same as ours. Hence this is another argument going in the favor of our proposed analysis.
> Here is the link of the code : https://github.com/anonymous-authors-2018/Generative_models_from_the_perspective_of_Continual_learning
> We modified Figure 2 and 3 for better readability.
>
>
> We added a summary of modification in a common answer on open review.

---

### Official Review · AnonReviewer3 · 2018-11-02
**Empirical analysis of CL is welcomed, but a few concerns with the experimental set-up.**

**Rating:** 5
**Confidence:** 3

**Review:**

This paper performs an empirical comparison of models and CL methods in a generative setting. The main motivation of the paper is to make statements about which model/method combinations are best to use for generative tasks in the CL setting. In short, the paper provides an empirical analysis and evaluation of the combination of CL methods and generative models.

The datasets used for comparison are MNIST, Fashion MNIST, and CIFAR10. For each dataset, sequential (class by class) generative tasks are introduced, aligning with the CL setting. The models investigated are VAEs, GANs, and WGANs, along with their (class) conditional counter-parts. The CL methods investigated are (i) fine-tuning (a simple baseline), (ii) rehearsal methods, (iii) elastic weight consolidation (EWC), and (iv) generative replay (GR). The authors propose to use two evaluation metrics: Fréchet Inception Distance (FID) measures the quality of the generated images, and fitting capacity (FC) measures the usefulness of the images to train classifiers.

Pros:
- The authors are correct in pointing out that most of the work on CL has been restricted to the discriminative case, and that there is value in exploring generative tasks in the CL setting.
- Empirical and experimental evaluation of this sort are useful, and help the community better understand the relationship between model, CL method, and task. Such an evaluation and in-depth analysis is welcomed in CL, especially in the generative setting.
- The authors draw a number of useful conclusions e.g., regarding the usefulness and dangers of employing the different CL methods.

Cons:
- My main concern with this paper regards the evaluation metrics used. The authors propose quality metrics for the generative model, both of which (directly or indirectly) measure the quality of the generated images. In this setting, it is unsurprising that GANs outperform VAEs, as they are known to generate higher-quality images. This however, does not necessarily mean that they are better at the continual learning task (i.e., avoiding catastrophic forgetting). It seems to me that one source from which to draw would be [1], which conducted a very rigorous and useful empirical evaluation of generative models, and the methodology followed there (i.e., evaluating marginal log-likelihoods via annealed importance sampling) would be more convincing evidence for empirical comparison of models, as it would somewhat detach the quality of the generated images from the ability of the model to avoid catastrophic forgetting.

Using their proposed image-quality metrics, the authors make statements such as: "Our results do not give a clear distinction between conditional and unconditional models. However, adversarial methods perform significantly better than variational methods. GANs variants are able to produce better, sharper quality and variety of samples, as observed in Fig. 13 and 14 in Appendix G. Hence, adversarial methods seem more viable for CL." My impression is that this statement on the viability of VAEs vs GANs for CL, which is a major point of the paper, does not follow from the empirical results on the quality of the generated images. It seems quite predictable that the GAN-based models would produce higher quality images, regardless of catastrophic forgetting.

Additional (minor) comments:
- Sec. 2 could consist of a more thorough review of the literature, with a more in-depth comparison of the different CL methods proposed and evaluated in the paper.
- Sec. 2 contains a number of statements of the form: "restricted to VAEs only". For many of the cases it is not immediately clear why this is true, and in my opinion the authors should either drop those comments, or make them rigorous.
- VCL "use specific weights for each task, which only works for the setting where the number of tasks is known in advance". Unclear what exactly this means or why this is true.
- "while the teacher retains knowledge" - how does it "retain knowledge", how is this then transferred to the student, and why is this restricted to VAEs?

Experimental protocol:
- Core-sets for the rehearsal as proposed by [1] could be an interesting extension. It is unclear how the samples were selected for rehearsal, and core-sets represent a principled way to do so, that would also be interesting to compare in this setting to a random baseline.
- For VAEs, a potentially better metric of their ability (other than the log-likelihood as suggested by [2]) would be fitting capacity (or other metric) over learned latent space rather than the reconstructed image-space.

Overall, my impression is that while an empirical analysis of CL methods in the generative setting is a useful concept, the submission in its current form requires some improvement. In particular, I am worried that the choice of evaluation metrics may lead to incorrect (or partially correct) conclusions, which could of course have a negative impact on the research into CL. It also seems that the paper could use some further polishing in both writing and presentation. As such, I encourage the authors to continue the work on this empirical analysis, and perhaps submit in again to future conferences.

[1] - Nguyen et al. Variational Continual Learning, ICLR 2018
[2] - Wu et al. On the Quantitative Analysis of Decoder-Based Generative Models, ICLR 2017

---

> ### Author Response · Authors · 2018-11-26
> **Answer**
>
> Dear Reviewer,
> Thanks for the review and the advices.
> We agree that our evaluation does not rigorously evaluate catastrophic forgetting of a particular model, in particular, because we evaluate at the end of the task and not all task long. In fine what we evaluate is the state of the generator after a fixed number of epochs. What we don't see in our results and that may append in particular for GAN based models is that at the beginning of the task they forget everything and relearn almost from scratch everything. Nevertheless, if a model A forgets everything and then learn to produce better samples than a model B that successfully made a smooth transition, model A is better for what we evaluate. We wanted to evaluate the ability to store knowledge and maintain it, not the smoothness of the transition (even if it may be very interesting when the time/computational power available to learn a new task is limited).
> We also agree that it is not very surprising that GANs outperform VAEs in our settings however no study compared them before. Furthermore, we do not claim to have found a model that learn continuously throughout the sequential tasks. Nevertheless, even if it is known that GANs generate good samples it is not straightforward to us to assume that the cumulated errors induced by training recurrently on generated samples will not produce poor samples (as it is shown for CIFAR10).
>
>
>
>
> - About evaluating marginal log-likelihoods via annealed importance sampling: Unfortunately, we did not have time to implement the evaluation of the log-likelihoods estimation  via annealed importance sampling
>
> - Concerning the evaluation of catastrophic forgetting without the consideration of absolute performance, Figure 4 now compares GAN and VAE.
>
> We modified the related work (Sec. 2) according to the comments provided to make it more clear.
> - About the approach restricted to VAE, it means that the approach presented has only been experimented on VAEs. We clarified this in the paper.
> - The samples for Rehearsal have been selected randomly. It is indeed a good idea to use core-sets, as it is a principled way to select the samples.
> - We clarified our statements on VCL.
> - When we say “retain knowledge” it just means that the model is able to maintain its performance from past tasks.
>
> “The authors propose quality metrics for the generative model, both of which (directly or indirectly) measure the quality of the generated images”
> The fitting capacity does not only evaluate the quality of samples but also if their variability fit the variability of the original dataset. Training a classifier with a few good samples will not achieve as good as training with various good samples. One could say that if the generator only learns discriminative features the classification task could be achieved easily and produce a high Fitting Capacity but first there is no reason that an unsupervised task learns only the exact features needed for a supervised task and secondly if it was the case the FID would be very high.
>
> Experimental protocol :
> the samples were selected randomly for rehearsal. Using an extended version of rehearsal could be an interesting extension to our work but at the moment we only concentrate on vanilla approaches for each Continual Learning category to get an insight of performance difference.
>
>
>  - About evaluating the latent space rather than the output space for VAE:
> In the same spirit as what has been previously said, in our study we wanted to compare different strategies and models in a sequence of generation tasks. This means that we want to evaluate first of all the samples produced by the model and see how they change task after task and not how weights or latent space evolve.
>
> We added a summary of modification in a common answer on open review.

---

### Official Review · AnonReviewer1 · 2018-11-02
**A comprehensive evaluation of existing methods lacking novelty and insight**

**Rating:** 4
**Confidence:** 4

**Review:**

This paper evaluates and compares various methods for learning GANs in a Continual Learning setting, i.e., only some of the classes are available during training. It evaluates different continual learning methods including rehearsal, EWC and generative replay applied to training several deep generative models, like GAN, CGAN, WGAN, WGAN-GP, VAE and CVAE on MNIST, Fashion MNIST and CIFAR. The authors conclude with these experimental results that generative replay is the most effective method for such a setting, and found it is difficult to generate CIFAR10 images that can be classified successfully by an image classifier.

I appreciate the authors for providing so much detailed experimental results to the community, but this paper lacks novelty in general. All the CL methods the authors evaluate come from other papers that are already using these methods for generative models: rehearsal has been used in VCL Nguyen et al. (2017), EWC comes directly from Seff et al. (2017), and generative replay has been used by Wu et al. (2018a). The authors also fail to provide any valuable insight with these experimental results, e.g., analyzing why generative replay fails to improve VAEs.

I expect to see more exciting results coming from the authors, but the paper is not mature enough for acceptance this time.

---

> ### Author Response · Authors · 2018-11-26
> **Answer**
>
> Dear Reviewer,
> Thanks for the review,
> We agree with the comment about the current lack of insights. In response, we added in paper comments and additional experiment about EWC, and why we believe it is bound to fail in our experimental setting. (Sec. 5.1.1). We also added comments about why specific combinations (VAE + Rehearsal and GAN + Generative Replay) perform well. (Sec. 5.1.2).
>
> We understand the need for novel approaches to make the state of the art progress. However, our goal was to extensively evaluate existing continual learning methods in common settings with various generative models. We did not want to be influenced/biased by a personal approach in the results we show.
> State of the art in continual learning propose many methods that are not always compared to each other. Neutral empirical studies like this one help to keep track of what exists and what is successful or not, and we think that such contribution should not be overlooked because no new approach is proposed here.
> Even if we did not produce a new approach, we have experimented a lot of combinations that have never been experimented and compared to each other in a common setting with a new way of evaluation for continual learning.
>
> We added a summary of modifications in a common answer on open review.

---

### Author Response · Authors · 2018-11-26
**Summary of modifications**

Here is a summary of the modifications we made to the paper:

We added several analyses and additional experiments in the Results section.
- More extensive experiments on CIFAR: we show that Rehearsal suffers from overfitting to the samples kept in memory, which was suggested by the results on MNIST and Fashion MNIST. Hence, we confirm this intuition thanks to the additional experiment. (Sec. 5.2)
- Comments and additional experiment about EWC, and why we believe it is bound to fail in our experimental setting. (Sec. 5.1.1)
- Comments about why specific combinations (VAE + Rehearsal and GAN + Generative Replay) perform well. (Sec. 5.1.2)
- Comments and figure (Fig. 4)  that explains how models suffer from forgetting. (Sec. 5.1.2)
We also added in appendix exact setting of Seff et al (2017) experiments ( Appendix G )
Furthermore, we added modifications to related work (Sec. 2) to make it more clear

---

### Meta-Review · Area_Chair1 · 2018-12-15

**Confidence:** 5
**Recommendation:** Reject

**Metareview:**

This paper presents empirical evaluation and comparison of different generative models (such as GANs and VAE) in the continual learning setting.
To avoid catastrophic forgetting, the following strategies are considered: rehearsal, regularization, generative replay and fine-tuning. The empirical evaluations are carried out using three datasets (MNIST, Fashion MNIST and CIFAR).

While all reviewers and AC acknowledge the importance and potential usefulness of studying and comparing different generative models in continual learning, they raised several important concerns that place this paper bellow the acceptance bar: (1) in an empirical study paper, an in-depth analysis and more insightful evaluations are required to better understand the benefits and shortcomings of the available models (R1 and R2), e.g. analyzing why generative replay fails to improve VAE, why is rehearsal better for likelihood models, and in general why certain combinations are more effective than others – see more suggestions in R1’s and R2’s comments. The authors discussed in their response to the reviews some of these questions, but a more detailed analysis is required to fully understand the benefits of this empirical study. (2) The evaluation is geared towards quality metrics for the generative models and lacks evaluation for catastrophic forgetting in continual learning (hence it favours GANs models) -- See R3’s suggestion how to improve.

To conclude, the reviewers and AC suggest that in its current state the manuscript is not ready for a publication. We hope the reviews are useful for improving and revising the paper.